# Cardiopulmonary Exercise Testing in the Age of New Heart Failure Therapies: Still a Powerful Tool?

**DOI:** 10.3390/biomedicines11082208

**Published:** 2023-08-06

**Authors:** Pedro Garcia Brás, António Valentim Gonçalves, João Ferreira Reis, Rita Ilhão Moreira, Tiago Pereira-da-Silva, Pedro Rio, Ana Teresa Timóteo, Sofia Silva, Rui M. Soares, Rui Cruz Ferreira

**Affiliations:** 1Cardiology Department, Hospital de Santa Marta, Centro Hospitalar Universitário de Lisboa Central, 1169-024 Lisbon, Portugal; 2NOVA Medical School, Faculdade de Ciências Médicas (NMS|FCM), 1169-056 Lisbon, Portugal

**Keywords:** heart failure with reduced ejection fraction, cardiorespiratory exercise testing, heart failure therapies, heart transplantation, peak oxygen consumption, VE/VCO_2_ slope

## Abstract

Background: New therapies with prognostic benefits have been recently introduced in heart failure with reduced ejection fraction (HFrEF) management. The aim of this study was to evaluate the prognostic power of current listing criteria for heart transplantation (HT) in an HFrEF cohort submitted to cardiopulmonary exercise testing (CPET) between 2009 and 2014 (group A) and between 2015 and 2018 (group B). Methods: Consecutive patients with HFrEF who underwent CPET were followed-up for cardiac death and urgent HT. Results: CPET was performed in 487 patients. The composite endpoint occurred in 19.4% of group A vs. 7.4% of group B in a 36-month follow-up. Peak VO_2_ (pVO_2_) and VE/VCO_2_ slope were the strongest independent predictors of mortality. International Society for Heart and Lung Transplantation (ISHLT) thresholds of pVO_2_ ≤ 12 mL/kg/min (≤14 if intolerant to β-blockers) and VE/VCO_2_ slope > 35 presented a similar and lower Youden index, respectively, in group B compared to group A, and a lower positive predictive value. pVO_2_ ≤ 10 mL/kg/min and VE/VCO_2_ slope > 40 outperformed the traditional cut-offs. An ischemic etiology subanalysis showed similar results. Conclusion: ISHLT thresholds showed a lower overall prognostic effectiveness in a contemporary HFrEF population. Novel parameters may be needed to improve risk stratification.

## 1. Introduction

In the past decade, the development of several pharmacological classes and innovative cardiovascular procedures in the treatment of heart failure with reduced ejection fraction (HFrEF) has led to a significant improvement in HF outcomes [1]. Sacubitril/valsartan was associated with a 20% reduction in both heart failure (HF) hospitalizations and cardiovascular mortality in comparison with enalapril [2]. Sodium-glucose cotransporter-2 inhibitors (SGLT2i) showed a significant prognostic benefit in HFrEF [3,4], and intravenous (IV) ferric carboxymaltose was associated with improved exercise capacity in patients with iron deficiency [5,6]. Cardiovascular procedures such as cardiac resynchronization therapy (CRT) [7], transcatheter edge-to-edge repair (TEER) in the management of functional mitral regurgitation [8], and catheter ablation of atrial fibrillation [9] have been increasingly used with promising results. Furthermore, the progressive streamlining of HF care, including intensive treatment strategies such as rapid up-titration of guideline-directed medication [10], improved transition care interventions, and the implementation of programs with close follow-up after an acute HF admission [11], have also contributed to a substantial reduction in HF events. These new drug classes and HF interventions have led to a change in the paradigm of HFrEF treatment and outcomes and thus risk stratification thresholds of HF patients in the last decades may not be as accurate in a contemporary HF population.

Cardiopulmonary exercise testing (CPET) is a key complementary exam in the risk stratification in patients with HFrEF, including in the decision for heart transplantation (HT) listing [12,13]. Peak oxygen consumption (pVO_2_) [14,15,16] and the minute ventilation-carbon dioxide production relationship (VE/VCO_2_ slope) [14,16,17] have been extensively studied in the risk stratification of patients with HFrEF, correlating with HF outcomes. According to the 2016 Society for Heart Lung Transplantation (ISHLT) listing criteria for HT [18], a cut-off for pVO_2_ of ≤12 mL/kg/min should be used to guide listing for HT in patients under β-blocker therapy [19], while a cut-off of ≤14 mL/kg/min should be used for patients intolerant of a β-blocker. In the presence of a sub-maximal CPET, the use of a VE/VCO_2_ slope of >35 and, in females and young patients (<50 years), a percent of predicted pVO_2_ ≤ 50% may be considered as determinants in listing for transplantation [18]. However, the evidence for these cut-off values stems from older trials (before 2010) [13,14,16,17] in which the aforementioned contemporary HF therapies were not available at the time.

There is limited evidence regarding the prognostic power of CPET parameters in a contemporary HFrEF population. CRT in patients with HFrEF to restore ventricular synchrony is associated with reduced cardiovascular outcomes and improved exercise capacity [7]. However, a substudy of the COMPANION trial has shown that CRT did not impact the predictive power of pVO_2_ on adverse cardiac events [20]. However, recent studies evaluating the predictability of pVO_2_ in HFrEF patients with a CRT device undergoing evaluation for HT suggested a cut-off of ≤10 mL/kg/min for risk stratification in the device era [21].

The aim of this study was to evaluate the prognostic difference in CPET parameters between an HFrEF cohort submitted to maximal CPET between 2009 and 2014 and an HFrEF cohort submitted to CPET between 2015 and 2018.

## 2. Materials and Methods

### 2.1. Study Population

The study included a retrospective single-center analysis from January 2009 to December 2018. During this period, we evaluated consecutive HFrEF patients with a left ventricular ejection fraction (LVEF) of ≤40% and in New York Heart Association (NYHA) class II or III who underwent CPET. All the patients were referred for assessment by the HF team and possible indication for HT or mechanical circulatory support (MCS).

### 2.2. Study Protocol

All patients’ clinical data were assessed including HF etiology (ischemic vs. non-ischemic), implanted cardiac devices, medication, comorbidities, NYHA class, laboratory, electrocardiographic, and echocardiographic data, CPET data, and Heart Failure Survival Score (HFSS) [22]. In patients under follow-up after 2015, in the event of hospitalization for heart failure, a clinical and laboratory assessment was routinely performed within 2 to 4 weeks by the HF team.

### 2.3. Exclusion Criteria

Age less than 18 years.Planned percutaneous coronary revascularization or cardiac surgery.Exercise-limiting comorbidities (cerebrovascular disease, musculoskeletal impairment, or severe peripheral vascular disease).Previous HT.Elective HT during the follow-up period.Submaximal CPET (defined as one with a peak RER of ≤1.05 [18]).Unavailable or missing clinical data on the composite endpoint.Lost to 36-month follow-up.

### 2.4. Cardiorespiratory Exercise Testing

A maximal symptom-limited treadmill CPET was performed using the modified Bruce protocol (GE Marquette Series 2000 treadmill, Marquette, Camarillo, CA, USA). Gas analysis was preceded by calibration of the equipment. VE, VO_2_, and VCO_2_ were acquired breath-by-breath with a SensorMedics Vmax 229 gas analyzer (SensorMedics Corp., Yorba Linda, CA, USA). Heart rate (HR) was measured by continuous ECG. Blood pressure (BP) was obtained manually with a sphygmomanometer, and oxygen saturation was monitored by pulse oximetry. Patients were encouraged to exercise until the target respiratory exchange ratio (RER) was reached. A maximal cardiopulmonary exercise test was defined according to the 2016 ISHLT guidelines [18] as one with an RER > 1.05. pVO_2_ was defined as the highest 30-s average achieved during exercise and was normalized for body mass. AT was determined by combining the standard methods (V-slope preferentially and ventilatory equivalents). The VE/VCO_2_ slope was calculated by least squares linear regression, using data acquired throughout the exercise. The cardiorespiratory optimal point (COP) was measured as the minimum value of the ventilatory equivalent for oxygen (minimum VE/VO_2_). The partial pressure of end-tidal carbon dioxide (PetCO_2_) was reported before exercise and at AT in mmHg units. Peak oxygen pulse was calculated by dividing derived pVO_2_ by maximum HR during exercise and was expressed in ml per beat. Circulatory power was calculated as the product of pVO_2_ and peak systolic blood pressure and ventilatory power was calculated by dividing peak systolic BP by the VE/VCO_2_ slope. HR reserve was calculated as the difference between the maximum HR achieved with exercise and resting HR. HR recovery in the first minute after exercise was defined as the difference between the maximum HR achieved with exercise and HR one minute into recovery. Several composite CPET parameters were also automatically calculated.

### 2.5. Follow-Up and Endpoint

All patients were followed for 36 months from the date of the CPET. The primary endpoint was a composite of cardiac death and urgent HT occurring during unplanned hospitalization for worsening inotrope-dependent HF. Data were obtained from outpatient clinic visits and a review of medical charts.

### 2.6. Statistical Analysis

All analyses compared patients who underwent CPET between January 2009 and December 2014 (Group A) and patients who underwent CPET between January 2015 and December 2018 (Group B). A subgroup analysis evaluating patients with ischemic HFrEF was performed. Statistical analysis was performed using the Statistical Package for the Social Sciences (SPSS), V.23.0 for Windows. Point estimates and 95% CI were presented for all mean estimates.

For categorical variables, descriptive statistics were presented as the absolute frequency (number) and relative frequency (percentage). Normally distributed continuous variables were reported as the mean (and standard deviation), and non-normally distributed continuous variables were presented as the median (and interquartile range [IQR]). Visual analysis of the histogram and the Kolmogorov–Smirnov test were used to test normality assumptions.

Comparisons between categorical variables were assessed with Pearson’s Chi-square test. Comparisons between continuous variables were evaluated with Student’s *t*-test (for normally distributed variables) or the Mann–Whitney U test (for non-normally distributed variables). 

A time-dependent Cox proportional hazards regression model was used to identify CPET parameters associated with HF events. Variables with a *p*-value < 0.200 in the univariate analysis were then considered in a multivariate model, adjusted for potential confounders, to determine independent predictors associated with HF outcomes and calculate adjusted hazard ratios (HR), in the total cohort and in each group. Results are reported as HR and 95% confidence interval (CI).

CPET parameters were further explored by receiver operating characteristic (ROC) curves to assess their sensitivity, specificity, and positive predictive value in predicting HF events according to the cut-off values recommended by the ISHLT guidelines [18]: pVO_2_ of ≤12 mL/Kg/min (pVO_2_ of ≤14 mL/Kg/min in patients intolerant of a β-blocker), VE/VCO_2_ slope of >35, and percent of predicted pVO_2_ of ≤50%. The DeLong et al. test [23] was used to assess the significance of the difference between the areas under the curves (AUC) from two ROC curves derived from each group. Additionally, the ROC curves were analyzed to determine the optimal cut-off values for predicting HF events. The Youden index (*J*) was used to identify the best cut-off value, combining the highest sum of sensitivity and specificity.

The Kaplan–Meier method estimated the event-free survival rate and a log-rank test rendered comparisons between groups according to the ISHLT guideline-recommended cut-off values for pVO_2_ and VE/VCO_2_ slope [18]. A significance level of α = 5% was considered whenever statistical hypothesis testing was used.

## 3. Results

### 3.1. Baseline Characteristics

A total of 487 patients were enrolled in the study, 283 in group A and 204 in group B (Figure 1). The baseline characteristics of both groups are presented in Table 1. The mean age was 58.6 ± 11.1 years, with 79% males, ischemic etiology in 57% of patients, 77% in NYHA class II and 23% in NYHA class III, and 25% in AF rhythm, with a mean LVEF of 29.8 ± 7.9%. Furthermore, 88% of patients were taking β-blockers and 77% were taking mineralocorticoid receptor antagonists (MRAs), with no significant difference between groups. In group A, 97% of patients were on angiotensin-converting-enzyme inhibitors (ACEi) or ARBs, while in group B, 56% were on an ACEi/ARB and 39% were taking sacubitril/valsartan. Moreover, 20% of patients in group B were on SGLT2i. In our cohort, 64% of patients had an ICD, out of which 23% had a CRT-D system, with no difference between groups. In addition, 16% of patients in group B were treated with IV ferric carboxymaltose, 11% were previously submitted to AF catheter ablation, and 8% to mitral TEER with a MitraClip^®^ system. These therapies were not available at the time for HF management in group A. The mean Heart Failure Survival Score (HFSS) was 8.6 ± 1.1, with no difference between groups.

All patients underwent CPET, with a mean respiratory exchange ratio (RER) of 1.13 ± 0.06. Patients in group B showed a lower pVO_2_ and a higher VE/VCO_2_ slope. The CPET parameters are shown in Table 1.

### 3.2. Primary Endpoint

In a 36-month follow-up, the primary endpoint occurred in 70 (14.4%) patients, with 52 patients experiencing cardiac death and 18 patients undergoing urgent HT (Table 2). No patients required urgent MCS. Patients in group B had a lower occurrence of the primary endpoint (7.4% vs. 19.4%, *p* < 0.001), driven by a lower cardiac death (4.9% vs. 14.8%, *p* < 0.001). There was no significant difference between groups regarding urgent HT, as shown in Table 2.

### 3.3. Relationship between Cardiopulmonary Exercise Test Prognostic Parameters and Primary Endpoint

In a multivariable time-dependent Cox regression model, pVO_2_ (adjusted HR 0.871, *p* < 0.001) and the percent of predicted pVO_2_ (adjusted HR 0.953, *p* < 0.001) were associated with the composite endpoint in a 36-month follow-up in both groups. VE/VCO_2_ slope (adjusted HR 1.027, *p* = 0.019) was also a predictor of the composite endpoint in both groups. The univariable and multivariable analysis is presented in Table 3. In this multivariable model adjusted for potential confounders, these associations were independent of body mass index, age, sex, estimated glomerular filtration rate, diabetes mellitus, and smoking. The majority of the remaining CPET parameters were not correlated with the composite endpoint in the multivariable analysis. The partial pressure of end-tidal carbon dioxide (PetCO_2_) at the anaerobic threshold (AT) was a predictor of outcomes in the total cohort, but not in the subgroup analysis. PetCO_2_ at rest was associated with the composite endpoint in group B with borderline statistical significance (Table 3).

In a ROC curve analysis (Table 4), pVO_2_ and VE/VCO_2_ were predictors of the composite endpoint in a 36-month follow-up in both groups. These associations were numerically superior in group A in comparison with group B, both for pVO_2_ (AUC 0.775 vs. AUC 0.732, *p* = 0.571) and VE/VCO_2_ slope (AUC 0.802 vs. AUC 0.752, *p* = 0.569). The ROC curves for both groups are illustrated in Figure 2. Likewise, the percent of predicted pVO_2_ correlated with the occurrence of the composite endpoint, both in group A (AUC 0.787, *p* < 0.001) and in group B (AUC 0.725, *p* = 0.004) (Appendix A). In a subgroup analysis evaluating patients with ischemic etiology, group A also showed a numerically superior AUC regarding pVO_2_ (0.752 vs. 0.691, *p* = 0.558), VE/VCO_2_ slope (0.803 vs. 0.771, *p* = 0.756), and percent of predicted pVO_2_ (0.764 vs. 0.580, *p* = 0.135). Although other CPET parameters, including O_2_ pulse, circulatory power, ventilatory power, COP, PetCO_2_ at rest, and PetCO_2_ at AT, were significant predictors of the composite endpoint, their predictive power was inferior to pVO_2_, the percent of predicted pVO_2_, or VE/VCO_2_ slope (Table 4).

### 3.4. ISHLT Cut-Offs for Heart Transplantation Listing

A total of 70 (14.4%) patients had a pVO_2_ ≤ 12 mL/kg/min (or ≤14 mL/kg/min if intolerant of a β-blocker) and patients below this value had a worse outcome (HR 2.351, 95% CI 1.359–4.068, *p* = 0.002). This pVO_2_ cut-off showed a positive predictive value (PPV) of 58.8% in group A and a PPV of 13.2% in group B for the composite endpoint in a 36-month follow-up, with a similar Youden index between groups (*J* 0.22 vs. *J* 0.22) (Table 5). A Kaplan–Meier survival analysis (Figure 3A) showed that the pVO_2_ ≤ 12 mL/kg/min (or ≤14 mL/kg/min if intolerant of a β-blocker) cut-off was a significant discriminator of the composite endpoint in group A (log-rank *p* < 0.001), but not in group B (log-rank *p* = 0.071). A subanalysis evaluating patients with ischemic HFrEF also showed a higher positive predictive value of the ISHLT-recommended pVO_2_ cut-off in group A compared to group B (PPV 60.0% vs. PPV 10.0%).

Regarding the VE/VCO_2_ slope, 166 (34.1%) patients had a value of >35. Likewise, patients above this value had a higher rate of the composite endpoint (HR 3.587, 95% CI 2.194–5.864, *p* < 0.001). In a 36-month follow-up, the VE/VCO_2_ slope cut-off of >35 presented a higher PPV in group A compared to group B (PPV 45.7% vs. 10.4%). Although this cut-off showed slightly lower sensitivity (58% vs. 67%), the Youden index was higher in group A compared to group B (*J* 0.41 vs. *J* 0.19) (Table 5). This VE/VCO_2_ slope cut-off was shown to be an accurate predictor of worse outcomes in group A (log-rank *p* < 0.001) in survival analysis (Figure 3B), but not in group B (log-rank *p* = 0.145). In an ischemic HFrEF subanalysis, the recommended VE/VCO_2_ slope threshold value presented a PPV of 57.1% in group A and a PPV of 10.7% in group B.

In our cohort, 143 (29.4%) patients had a percent of predicted pVO_2_ of ≤50%. This cut-off was correlated with the composite outcome (HR 3.560, 95% CI 2.219–5.710, *p* < 0.001). The percent of predicted pVO_2_ ≤50% threshold showed a PPV of 48.3% in group A and a PPV of 11.8% in group B for the composite endpoint in a 36-month follow-up, with a sensitivity of 50.9% in group A and 66.6% in group B. Moreover, this percent of predicted pVO_2_ threshold showed a higher Youden index in group A compared to group B (*J* 0.38 vs. *J* 0.27). A survival curve analysis (Appendix A) showed that this cut-off was an accurate discriminator of the composite endpoint only in group A (*p* < 0.001) (group B *p* = 0.059). An ischemic HFrEF subanalysis also revealed a higher PPV of the percent of predicted pVO_2_ ≤50% threshold value in group A compared to group B (PPV 69.6% vs. PPV 8.3%).

### 3.5. Alternative pVO_2_ and VE/VCO_2_ Slope Cut-Offs

In a subsequent analysis evaluating alternative cut-off values in group B, a pVO_2_ ≤ 10 mL/kg/min yielded a higher PPV of 29.4% for the composite endpoint. A Kaplan–Meier analysis (Figure 4A) confirmed the discriminative power of this cut-off (log-rank *p* < 0.001). Although the sensitivity associated with this cut-off was low (<50%), the Youden index was higher compared to the pVO_2_ ≤ 12 mL/kg/min threshold (*J* 0.30 vs. *J* 0.22). In patients with ischemic HFrEF, this cut-off showed a PPV of 22.2%.

A proposed VE/VCO_2_ slope threshold of >40 showed a specificity of 81% and a PPV of 19.5% for the composite outcome in group B, albeit with a slightly lower sensitivity (62%). A VE/VCO_2_ slope > 40 showed a higher Youden index than the VE/VCO_2_ slope > 35 threshold (*J* 0.43 vs. *J* 0.19). This VE/VCO_2_ slope cut-off was also shown to be an accurate predictor of worse outcomes in survival analysis (log-rank *p* = 0.001) (Figure 4B). Similarly, in patients with HFrEF of ischemic etiology, the PPV associated with a VE/VCO_2_ slope > 40 was 22.7%.

A percent of predicted pVO_2_ of ≤40% was also associated with a higher PPV (20.0%) in group B and significant discriminative power in survival analysis (log-rank *p* = 0.006), with a similar Youden index compared to the percent of predicted pVO_2_ ≤ 50% cut-off (*J* 0.28 vs. *J* 0.27). The subanalysis evaluating patients with ischemic HFrEF showed a lower PPV of 12.5% in this subgroup for the ≤40% threshold.

## 4. Discussion

The main finding of our study was that, in an HFrEF cohort comprising primarily patients with ischemic HFrEF with regular follow-up by a dedicated HF team, optimized contemporary HF medical therapy (including sacubitril/valsartan), and cardiovascular procedures such as ICD/CRT implantation, AF catheter ablation, mitral TEER, or IV ferric carboxymaltose, the ISHLT guideline-recommended cut-off values for pVO_2_ (≤12 mL/kg/min, or ≤14 mL/kg/min if intolerant of a β-blocker) and VE/VCO_2_ slope (>35) showed similar and lower overall prognostic effectiveness, respectively, for major HF outcomes compared to a control group who underwent CPET from 2009 to 2014, when these therapies were not available. Furthermore, our study evaluated the prognostic effectiveness of several CPET parameters and proposed alternative thresholds for pVO_2_, the VE/VCO_2_ slope, and the percent of predicted pVO_2_, which may contribute to accurate risk stratification and patient selection for HT listing in a contemporary HFrEF population.

In group B, which is representative of a contemporary HFrEF population with broader medical and procedural management options, the composite endpoint of cardiac death or urgent HT was significantly reduced compared to group A, comprising patients who underwent CPET from 2009 to 2014. The increasing role of optimized medical therapy, with the introduction of novel drug classes, has led to a new paradigm in HFrEF management [24]. Notably, sacubitril/valsartan was associated with a reduction in major cardiac endpoints, including cardiovascular death and HF hospitalizations, in comparison to enalapril, as shown in the PARADIGM-HF trial [2]. The high percentage of patients in group B under treatment with sacubitril/valsartan contributed to the lower rate of cardiac outcomes in this group. Moreover, group B included patients under SGLT2i [3,4], patients submitted to AF catheter ablation [9], mitral TEER [8], and IV ferric carboxymaltose [5,6]. These interventions may have also contributed to the markedly improved cardiovascular outcomes in this group. There were no statistical differences between groups in the baseline percentage of patients with an ICD and patients with a CRT. However, these were numerically superior in group B, which may also have contributed to lower cardiac death. Importantly, the increasing regular follow-up by a dedicated HF team, implementing rapid up-titration of medical therapy after diagnosis [10], cardiac rehabilitation programs [25], improved transitional care interventions [11,26], and close follow-up after an HF hospitalization or an urgent visit for IV diuretics have also led to a reduction in adverse cardiac events.

Exercise intolerance is a fundamental manifestation of HF. CPET allows for a precise definition of maximum exercise capacity through the measurement of pVO_2_ and thus has an essential role in the risk stratification for advanced HF interventions including HT and ventricular assist devices [27]. However, according to the ISHLT guidelines [18], listing patients for HT based solely on the criterion of a pVO_2_ measurement should not be performed. The evaluation of candidates for listing should always be based on an integrated approach encompassing clinical variables, HF prognosis scores, CPET parameters, right-heart catheterization, and a comprehensive assessment of comorbidities. Nevertheless, CPET remains a cornerstone of patient selection for HT.

In our study, the CPET parameters pVO_2_, VE/VCO_2_, and percent of predicted VO_2_ were predictors of cardiovascular outcomes in a multivariable analysis in both groups, which is in keeping with their proven role in risk stratification in HFrEF [15,18,28]. However, these CPET variables all showed a numerically lower AUC in group B compared to group A, indicating that their predictive power may be weaker in a contemporary HF population. The remaining CPET parameters presented a lower predictive power than pVO_2_ or the VE/VCO_2_ slope.

Several studies have shown an association of different CPET variables with prognostic power for HF events. COP has been proposed as an alternative parameter for risk stratification in HF, particularly in patients with low-level exercise [29,30,31]. PetCO_2_, measured at rest, AT, and peak exercise, is associated with HF severity and has shown prognostic value in HFrEF [30,32]. In our cohort, PetCO_2_ at rest was an independent predictor of the composite endpoint in group B, although with a numerically lower AUC than either pVO_2_ or the VE/VCO_2_ slope. Despite increasing evidence on different CPET parameters, the ISHLT guidelines still recommend pVO_2_ and the VE/VCO_2_ slope in patient selection for HT listing [18].

Prognostic risk scores may also contribute to improved risk stratification in HFrEF and discriminating patients for HT listing. According to the ISHLT recommendations, patients with an HFSS [22] in the high-risk to medium-risk range or a Seattle Heart Failure Model (SHFM) [33,34] of <80% may be considered for HT listing [18]. A trial by Goda et al. showed that combining both prognostic risk scores outperformed either risk score alone in predicting cardiac events in patients undergoing HT evaluation [35], and the HFSS showed a higher predictive power than pVO_2_ to discriminate between patients at low or medium risk of death at 12 months [21].

A combination of pVO_2_ with prognostic risk scores may also assist in guiding HT listing, as a trial by Lewy et al. showed that the SHFM and pVO_2_ work synergistically to improve prognostic accuracy in patients with an intermediate risk of needing urgent HT [36]. Other prognostic risk models, such as Meta-Analysis Global Group in Chronic Heart Failure (MAGGIC) [37,38], Organized Program to Initiate Lifesaving Treatment in Hospitalized patients with HF (OPTIMIZE-HF) [39], and Get With The Guidelines-Heart Failure (GWTG-HF) [40] were associated with HT waiting-list mortality [41].

### 4.1. ISHLT Recommended Cut-Off Values

The ISHLT-recommended threshold of pVO_2_ ≤ 12 mL/kg/min (≤14 mL/kg/min if intolerant of a β-blocker) had a similar overall prognostic effectiveness in both groups while the VE/VCO_2_ slope > 35 and percent of predicted pVO_2_ ≤ 50% thresholds showed lower overall effectiveness in group B, indicating that these cut-off values may not accurately measure the risk for HF events in patients with optimized contemporary HF management strategies. In contrast, in the control group who underwent CPET from 2009 to 2014, these cut-offs presented higher prognostic effectiveness, in keeping with previous evidence. Indeed, the evidence for the definition of the ISHLT cut-offs stems from earlier trials [13,14,16,17], in which HFrEF treatment did not encompass the range of therapies that are currently available, from breakthroughs in medical therapy to cardiovascular procedures. 

A study by Paolillo et al. [42] comparing patients with HFrEF between 1993 and 2015 evaluated how the prognostic threshold of pVO_2_ and the VE/VCO_2_ slope has changed over the last 20 years in parallel with HFrEF prognosis improvement. The pVO_2_ and VE/VCO_2_ slope cut-offs able to identify a definite risk progressively decreased and increased over time, respectively. Thus, the authors highlighted that the prognostic threshold of pVO_2_ and the VE/VCO_2_ slope should be updated whenever the HFrEF prognosis improves.

A pVO_2_ ≤ 10 mL/kg/min has previously been associated with a higher risk for adverse outcomes in HF patients [43,44]. A trial by Corrà et al. compared the prognostic significance of pVO_2_ ≤10 mL/kg/min in an HfrEF cohort between 1994 and 1999 and a cohort between 2001 and 2008 in light of changes in medical treatment and management. The two-year outcomes of HF patients with pVO_2_ ≤ 10 mL/kg/min were significantly improved in the post-2000 era, suggesting that the indication for HT should not rely on a single CPET parameter, but rather on a multifactorial clinical approach [45].

A study by Goda et al. evaluated the prognostic accuracy of a pVO_2_ cut-off of ≤10 mL/kg/min in HfrEF patients with an ICD, CRT, or both (CRT-D) referred for HT. This cut-off was a strong predictor of HF outcomes, and the authors conclude that a pVO_2_ of ≤10 mL/kg/min rather than the traditional cut-off value may be more useful for risk stratification in the device era [21]. Nevertheless, there are conflicting data [46] and, in the latest guidelines, the ISHLT recommends maintaining the currently accepted pVO_2_ values [18].

Cattadori et al. [47] assessed the prognostic role of pVO_2_ in the era of beta-blockers and concluded that pVO_2_ maintains prognostic value independently of the presence of β-blocker therapy in severe HF patients. In addition, the authors confirmed the value of the pVO_2_ criteria for HT selection but with a lower cut-off (≤8 mL/kg/min) than the traditional threshold.

Furthermore, several trials have evaluated new thresholds for the VE/VCO_2_ slope in HFrEF, showing that the risk of HF events is continuous across a wide range of VE/VCO_2_ slope values [48,49]. A trial by Ferreira et al. showed that in a multicenter HFrEF population, a VE/VCO_2_ slope of ≥43 yielded the best discriminative power to predict the occurrence of death or HT in a 3-year follow-up [50]. In a recent trial from our center [51], a VE/VCO_2_ slope of ≥39 yielded a high specificity of 97% and discriminated high-risk from low-risk patients in a 60-month follow-up, outperforming pVO_2_ cut-offs of ≤10 or ≤12 mL/kg/min. In our cohort, in a contemporary HF population represented by group B, a pVO_2_ of ≤10 mL/kg/min and a VE/VCO_2_ slope of >40 showed a higher overall effectiveness to identify high-risk patients in a 36-month follow-up. A percent of predicted pVO_2_ of ≤40% was also associated with higher prognostic effectiveness in this group. These thresholds may provide an alternative dichotomous risk stratification tool, contributing in the integrated approach to identify high-risk HFrEF patients who may benefit from HT in the current age of HF medical therapy and cardiovascular procedures.

Although these proposed thresholds outperformed the traditional pVO_2_, VE/VCO_2_ slope, and percent of predicted pVO_2_ cut-off values, their overall prognostic effectiveness was low (*J* < 0.45). Likewise, these cut-off PPV values were in the 20–30% range, which was much lower than the PPV of the traditional thresholds in the control group (45–60% range). The sensitivity of the proposed thresholds was also low. These findings suggest that new variables may be necessary for risk stratification and patient selection for advanced HF therapies in a contemporary HFrEF population, which presents a challenge due to the increasingly wide variety of HF therapies, which are rapidly evolving, such as with the introduction of guanylate cyclase stimulators or selective cardiac myosin activators [52,53].

### 4.2. Study Limitations

There are limitations to our study that should be acknowledged. Firstly, this was a retrospective study with a relatively small sample enrolling patients in a single center and our results require confirmation in multiple centers for appropriate validation. Moreover, 79% of the enrolled patients were male.

Secondly, although our study included unmatched cohorts, patients were enrolled consecutively, mitigating the lack of randomization. In addition, the majority of the baseline characteristics were similar between groups, including age, gender, chronic kidney disease, NYHA class, and a similar percentage of ischemic HFrEF etiology, mitral regurgitation, right ventricular dysfunction, and ICD/CRT implantation. Notably, the HFSS, a current tool in the prognostic risk stratification of patients with HFrEF, did not significantly differ between groups. Medical therapy with neurohormonal antagonists, including β-blockers and mineralocorticoid receptor antagonists, was also high in both groups. As patients in group B were enrolled between 2015 and 2018, SGLT2i were not yet included as optimized standard-of-care therapy for non-diabetic patients in accordance with the recommendations at the time [54], with only 20% of the study population on SGLT2i. With the current evidence of this class’s effect on the reduction of major cardiovascular events in HFrEF [3,4], a higher percentage of patients on SGLT2i should be included in future trials. Sacubitril/valsartan was not yet available for patients enrolled in our study in group B in 2015, it was not always tolerated due to symptomatic hypotension, and it presents a higher cost than ACEi/ARBs. Thus, only over one-third of HFrEF patients in our study were under angiotensin receptor/neprilysin inhibitor therapy. Likewise, a higher rate of patients under sacubitril/valsartan should be enrolled in future trials. In addition, novel therapies such as guanylate cyclase stimulators or selective cardiac myosin activators were not available at the time of enrollment.

Thirdly, the proposed pVO_2_ cut-off of ≤10 mL/kg/min was only evaluated in patients taking β-blockers. As most patients (95%) in group B were taking β-blockers, our study did not have statistical power to infer a new pVO_2_ cut-off for patients intolerant of a β-blocker and thus the proposed threshold of pVO_2_ ≤ 10 mL/kg/min may not be applicable to this subset of HFrEF patients. 

Only patients with a maximal CPET were enrolled. Currently, there is no consensus on the optimal peak RER cut-off for the definition of maximal effort, particularly in HFrEF patients. Several cut-offs have been proposed, from 1.0 to 1.10 [12,27,30,32]. In our study, as the risk stratification for HT listing was being evaluated, we defined maximal CPET as one with a peak RER of 1.05 according to the ISHLT guidelines [18]. As patients with a submaximal CPET were excluded, our results and proposed thresholds may not be applicable to a general HFrEF population, particularly as pVO_2_ may present a lower prognostic value in patients with submaximal exercise capacity [55]. In this context, VE/VCO_2_ slope and percent of predicted pVO_2_ are pivotal parameters for risk stratification [18,27,28]. 

Moreover, our retrospective study evaluated patients with HFrEF using the same modified Bruce protocol, with variable total CPET duration. Personalization of the CPET protocol is crucial to accurately measure peak exercise in patients with HFrEF [56] by adjusting the magnitude of the work rate increment according to the patient’s cardiorespiratory status [57,58]. Future trials should incorporate personalized ramp exercise protocols with a defined duration to achieve reproducible results regarding peak exercise.

Finally, HF severity may have been lower in our cohort compared to other studies [48], as LVEF and pVO_2_ were higher and there was a lower rate of events, particularly HT. This should also be considered in the reproducibility of our findings. All the enrolled patients were referred to the HF team for an evaluation of the indication for advanced HF therapies, and thus our cohort may not be representative of a general HFrEF population including older patients or with several comorbidities.

## 5. Conclusions

In an HFrEF cohort under therapy with recent HF therapies, there was a significant reduction in the composite endpoint of cardiac death or urgent HT. ISHLT guideline-established cut-off values for pVO_2_ and the VE/VCO_2_ slope had similar and lower overall prognostic effectiveness, respectively. While the pVO_2_ ≤ 10 mL/kg/min and VE/VCO_2_ slope > 40 thresholds outperformed the traditional cut-offs, their overall prognostic effectiveness was low, suggesting that in the age of contemporary HFrEF therapies, new parameters for risk stratification may be needed for the timely identification of patients who may benefit from advanced HF therapies.

## Figures and Tables

**Figure 1 biomedicines-11-02208-f001:**
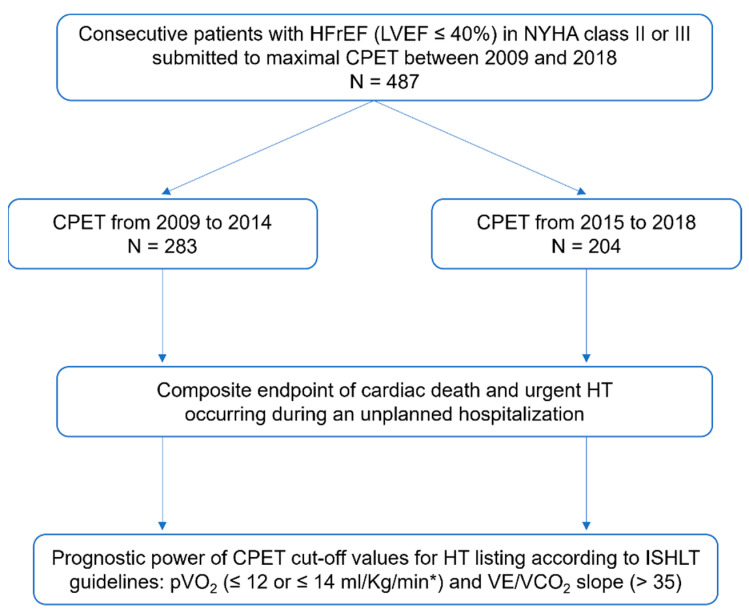
Study population flowchart. * in patients intolerant of a β-blocker. CPET: Cardiopulmonary exercise test; HFrEF: Heart failure with reduced ejection fraction; HT: Heart transplantation; ISHLT: International Society for Heart and Lung Transplantation; LVEF: Left ventricular ejection fraction; NYHA: New York Heart Association; pVO_2_: Peak oxygen consumption; VE/VCO_2_ slope: Minute ventilation-carbon dioxide production relationship.

**Figure 2 biomedicines-11-02208-f002:**
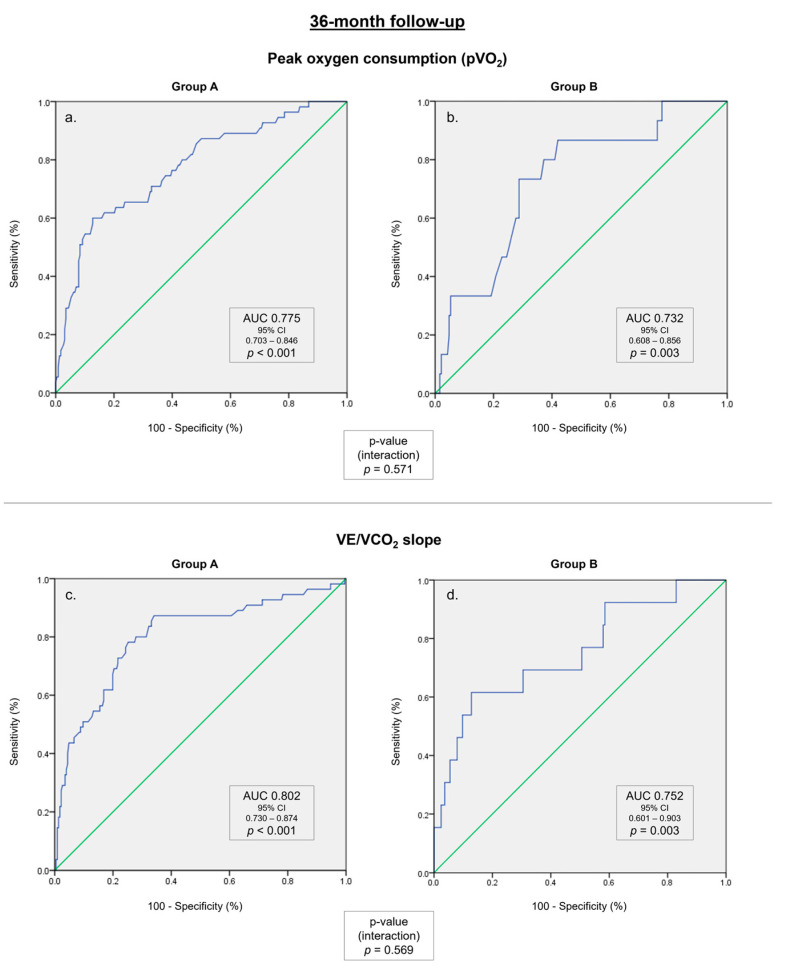
ROC curves for the composite endpoint in a 36-month follow-up. (**a**) Peak oxygen consumption (pVO_2_) in group A. (**b**) pVO_2_ in group B. (**c**) Minute ventilation-carbon dioxide production relationship (VE/VCO_2_ slope) in group A. (**d**) VE/VCO_2_ slope in group B.

**Figure 3 biomedicines-11-02208-f003:**
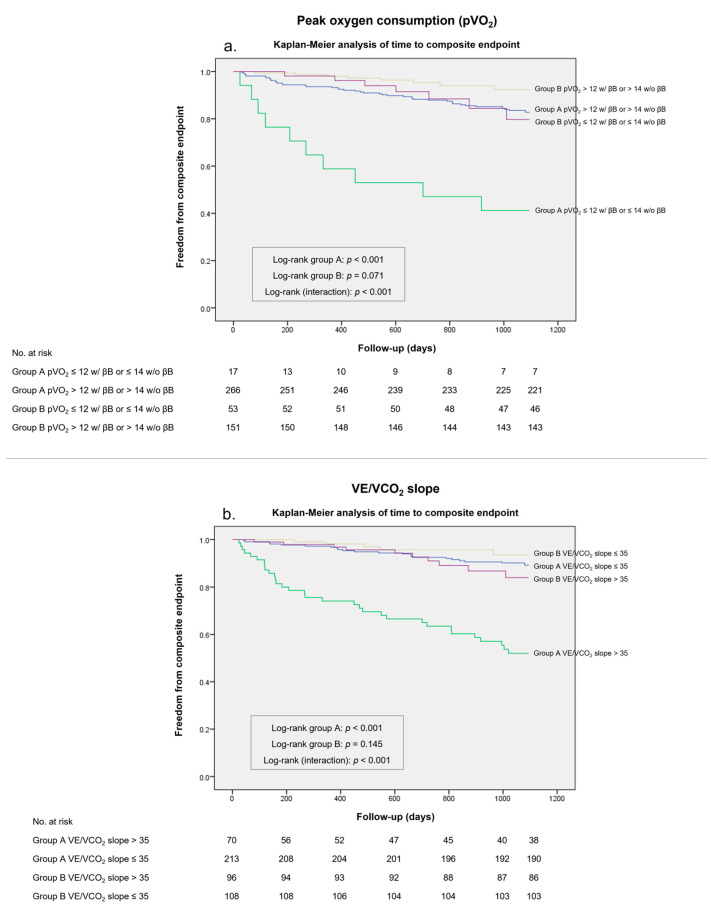
Kaplan–Meier survival analysis for the composite endpoint in a 36-month follow-up stratified according to the International Society for Heart and Lung Transplantation (ISHLT) guidelines in group A and group B. (**a**) Peak oxygen consumption (pVO_2_) of ≤12 mL/Kg/min (≤14 mL/kg/min if intolerant of a β-blocker [βB]). (**b**) Minute ventilation-carbon dioxide production relationship (VE/VCO_2_ slope) of >35.

**Figure 4 biomedicines-11-02208-f004:**
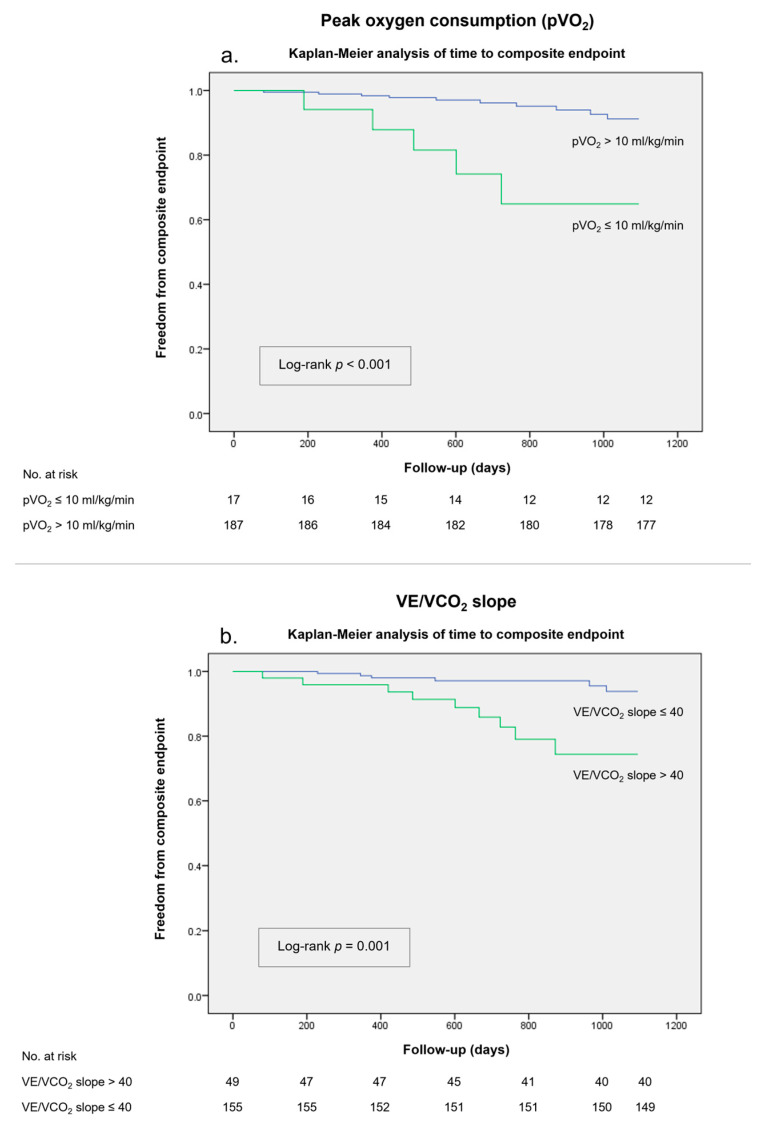
Kaplan–Meier survival analysis for the composite endpoint in a 36-month follow-up in group B stratified according to (**a**) peak oxygen consumption (pVO_2_) of ≤10 mL/Kg/min. (**b**) Minute ventilation-carbon dioxide production relationship (VE/VCO_2_ slope) of >40.

**Table 1 biomedicines-11-02208-t001:** Baseline characteristics of the study population.

	Overall (*n* = 487) *n*, %	Group A (*n* = 283) *n*, %	Group B (*n* = 204) *n*, %	*p*-Value
Clinical and demographic data				
Age (years)	56 ± 13	54 ± 12	58 ± 13	0.101
Male sex (*n*, %)	385 (79)	215 (76)	170 (83)	0.055
Body mass index (kg/m^2^)	27.1 ± 4.3	26.9 ± 4.2	27.3 ± 4.4	0.371
Ischemic etiology (*n*, %)	276 (57)	167 (59)	109 (53)	0.122
ACEi/ARB (*n*, %)	387 (79)	272 (97)	115 (56)	<0.001
ARNI (*n*, %)	80 (16)	0 (0)	80 (39)	<0.001
β-blocker (*n*, %)	430 (88)	247 (87)	183 (95)	0.060
MRA (*n*, %)	373 (77)	214 (76)	159 (82)	0.089
iSGLT2 (*n*, %)	41 (8)	0 (0)	41 (20)	<0.001
Digoxin (*n*, %)	134 (28)	108 (38)	26 (13)	<0.001
Diabetes (*n*, %)	108 (22)	59 (21)	49 (24)	0.659
CKD (*n*, %)	156 (32)	102 (36)	54 (26)	0.110
AF (*n*, %)	122 (25)	54 (19)	68 (33)	<0.001
AF catheter ablation (*n*, %)	23 (5)	0 (0)	23 (11)	<0.001
ICD * (*n*, %)	310 (64)	173 (61)	137 (67)	0.121
CRT (*n*, %)	112 (23)	59 (21)	53 (26)	0.323
Mitral TEER (*n*, %)	16 (3)	0 (0)	16 (8)	<0.001
IV ferric carboxymaltose (*n*, %)	32 (7)	0 (0)	32 (16)	<0.001
NYHA class II	375 (77)	221 (78)	151 (74)	0.802
NYHA class III	112 (23)	62 (22)	53 (26)	0.802
HFSS	8.6 ± 1.1	8.7 ± 1.0	8.5 ± 1.3	0.494
Laboratory data				
Creatinine, mg/dL	1.1 ± 0.4	1.1 ± 0.3	1.1 ± 0.4	0.656
eGFR, mL/min/1.73 m^2^	75.1 ± 29.0	74.2 ± 30.0	76.4 ± 27.6	0.415
Sodium, mEq/L	138.0 ± 3.0	137.3 ± 3.2	138.9 ± 2.5	<0.001
NT-proBNP, pg/mL	2202.5 ± 2101.7	2275.3 ± 2189.8	1815.9 ± 1643.2	0.312
Echocardiographic data				
LVEDD, mm/m^2^	64.1 ± 10.4	67.7 ± 9.8	62.6 ± 13.9	0.100
LVEF, %	29.8 ± 7.9	29.0 ± 7.5	31.3 ± 7.9	0.213
MR III-IV, %	72 (15)	42 (15)	30 (15)	0.257
RV dysfunction (*n*, %)	60 (12)	40 (14)	20 (10)	0.638
CPET parameters				
CPET duration	9.7 ± 4.4	10.4 ± 4.2	8.6 ± 4.5	<0.001
Peak RER	1.13 ± 0.06	1.14 ± 0.07	1.12 ± 0.05	0.003
Delta HR during exercise	49 [35–68] *	53 [39–70]	45 [31–61]	<0.001
HHR1	17 [10–26]	17 [11–26]	16 [9–27]	0.120
pVO_2_, mL/kg/min	18.3 ± 5.9	19.7 ± 5.7	16.4 ± 6.1	<0.001
VE/VCO_2_ slope	33.9 ± 9.6	31.3 ± 7.4	35.3 ± 9.8	<0.001
pVO_2_, mL/kg/min at AT	13.6 ± 4.6	15.5 ± 4.2	13.3 ± 4.6	0.048
O_2_ pulse, mL/kg/beat	0.14 ± 0.06	0.14 ± 0.04	0.13 ± 0.08	0.063
Circulatory power, mmHg·mL/kg/min	2798 ± 1550	3069 ± 1211	2420 ± 1862	<0.001
Ventilatory power, mmHg	4.8 ± 1.6	5.2 ± 1.6	4.1 ± 1.4	<0.001
COP	29.4 ± 7.4	29.2 ± 8.0	29.4 ± 7.4	0.940
PetCO_2_ at rest, mmHg	33.6 ± 4.9	33.1 ± 4.4	34.4 ± 5.3	0.007
PetCO_2_ at AT, mmHg	36.8 ± 6.0	36.8 ± 6.1	36.7 ± 5.9	0.846

* including CRT-D. Values are mean ± standard deviation or median [interquartile range]. *p* values are calculated by Student’s t test for independent samples or the Mann–Whitney U test as appropriate. ACEi: Angiotensin-converting enzyme inhibitors; AF: Atrial fibrillation; ARB: Angiotensin receptor blockers; ARNI: Angiotensin receptor neprilysin inhibitors; AT: Anaerobic threshold; BB: Beta-blockers; BMI: Body mass index; CKD: Chronic kidney disease; COP: Cardiorespiratory optimal point; CPET: Cardiopulmonary exercise test; CRT: Cardiac resynchronization therapy; HFSS: Heart Failure Survival Score; HRR1: Heart rate recovery in the first minute after finishing CPET; ICD: Implantable cardioverter-defibrillator; LVEF: Left ventricular ejection fraction; LVEDD: Left ventricular end-diastolic diameter; MRA: Mineralocorticoid receptor antagonists; MR: Mitral regurgitation; PetCO_2_: Partial pressure of end-tidal carbon dioxide; pVO_2_: Peak oxygen consumption; RER: Respiratory exchange ratio; RV: Right ventricular; TEER: Transcatheter edge-to-edge repair; VE/VCO_2_ slope: Minute ventilation-carbon dioxide production relationship.

**Table 2 biomedicines-11-02208-t002:** Adverse events at 36-month follow-up.

	Overall (*n* = 487) *n*, %	Group A (*n* = 283) *n*, %	Group B (*n* = 204) *n*, %	*p*-Value
Clinical and demographic data				
Combined primary endpoint	70 (14.4%)	55 (19.4%)	15 (7.4%)	<0.001
Total mortality	61 (12.5%)	48 (17.0%)	13 (6.4%)	<0.001
Cardiac mortality	52 (10.7%)	42 (14.8%)	10 (4.9%)	<0.001
Sudden cardiac death	19 (3.9%)	16 (5.6%)	3 (1.5%)	0.018
Death from worsening HF	33 (6.8%)	26 (9.2%)	7 (3.4%)	0.013
Urgent HT	18 (3.7%)	13 (4.6%)	5 (2.5%)	0.216

HF: heart failure; HT: heart transplantation.

**Table 3 biomedicines-11-02208-t003:** Clinical and cardiopulmonary exercise testing parameters—univariable and multivariable analysis for predicting the composite endpoint in a 36-month follow-up.

Total Cohort						
Model	Univariable HR	95% CI	*p*-Value	Multivariable HR	95% CI	*p*-Value
Age	1.002	0.983 to 1.020	0.865			
Male gender	1.451	0.762 to 2.762	0.257			
BMI	0.955	0.900 to 1.012	0.121	0.983	0.918 to 1.051	0.610
LVEF	0.924	0.879 to 0.952	<0.001	0.934	0.905 to 0.964	<0.001
eGFR	0.979	0.970 to 0.989	<0.001	0.986	0.975 to 0.996	0.005
Diabetes	1.148	0.637 to 2.070	0.871			
Smoker	1.604	0.986 to 2.610	0.225			
Peak VO_2_	0.879	0.840 to 0.920	<0.001	0.871	0.810 to 0.937	<0.001
Percent of predicted pVO_2_	0.960	0.948 to 0.973	<0.001	0.953	0.934 to 0.972	<0.001
VE/VCO_2_ slope	1.058	1.041 to 1.075	<0.001	1.027	1.005 to 1.051	0.019
Peak VO_2_ at AT, mL/kg/min	0.857	0.740 to 0.993	0.040	0.992	0.727 to 1.353	0.959
O_2_ pulse, mL/kg/beat	0.883	0.818 to 0.953	0.001	0.963	0.880 to 1.056	0.429
Circulatory power, mmHg·mL/kg/min	0.999	0.999 to 0.999	<0.001	0.999	0.999 to 1.000	0.075
Ventilatory power, mmHg	0.575	0.483 to 0.684	<0.001	0.775	0.564 to 1.011	0.755
COP	1.094	1.035 to 1.156	0.002	1.070	0.975 to 1.175	0.155
PetCO_2_ at rest, mmHg	0.895	0.849 to 0.943	<0.001	0.945	0.892 to 1.001	0.056
PetCO_2_ at AT, mmHg	0.864	0.827 to 0.901	<0.001	0.931	0.872 to 0.994	0.032
Group A						
Model	Univariable HR	95% CI	*p*-value	Multivariable HR	95% CI	*p*-value
Age	1.004	0.982 to 1.026	0.738			
Male gender	1.662	0.814 to 3.396	0.163	1.371	0.627 to 2.999	0.429
BMI	0.978	0.916 to 1.045	0.515			
LVEF	0.929	0.898 to 0.961	<0.001	0.954	0.920 to 0.989	0.010
eGFR	0.985	0.974 to 0.995	0.004	0.995	0.985 to 1.005	0.333
Diabetes	1.025	0.538 to 1.950	0.941			
Smoker	1.737	1.007 to 2.997	0.047	1.018	0.545 to 1.902	0.956
Peak VO_2_	0.804	0.752 to 0.860	<0.001	0.862	0.794 to 0.936	<0.001
Percent of predicted pVO_2_	0.937	0.920 to 0.954	<0.001	0.949	0.926 to 0.973	<0.001
VE/VCO_2_ slope	1.103	1.078 to 1.128	<0.001	1.062	1.026 to 1.099	0.001
Peak VO_2_ at AT, mL/kg/min	0.907	0.677 to 1.215	0.513	0.896	0.648 to 1.239	0.507
O_2_ pulse, mL/kg/beat	0.857	0.779 to 0.994	0.002	0.928	0.831 to 1.037	0.188
Circulatory power, mmHg·mL/kg/min	0.999	0.999 to 0.999	<0.001	0.999	0.999 to 1.000	0.440
Ventilatory power, mmHg	0.513	0.421 to 0.625	<0.001	0.844	0.593 to 1.201	0.346
COP	1.153	0.760 to 1.748	0.503	1.114	0.623 to 1.996	0.715
PetCO_2_ at rest, mmHg	0.918	0.860 to 0.980	0.010	0.962	0.903 to 1.027	0.251
PetCO_2_ at AT, mmHg	0.866	0.828 to 0.907	<0.001	0.984	0.924 to 1.047	0.602
Group B						
Model	Univariable HR	95% CI	*p*-value	Multivariable HR	95% CI	*p*-value
Age	1.015	0.974 to 1.058	0.485			
Male gender	1.172	0.264 to 5.197	0.834			
BMI	0.883	0.777 to 1.004	0.057	0.870	0.703 to 1.078	0.204
LVEF	0.933	0.873 to 0.997	0.041	0.946	0.857 to 1.045	0.274
eGFR	0.957	0.933 to 0.982	0.001	0.957	0.931 to 0.995	0.026
Diabetes	1.873	0.419 to 2.389	0.412			
Smoker	1.775	0.594 to 5.303	0.304			
Peak VO_2_	0.951	0.760 to 0.952	0.005	0.951	0.589 to 0.952	0.018
Percent of predicted pVO_2_	0.953	0.923 to 0.983	0.003	0.936	0.882 to 0.992	0.027
VE/VCO_2_ slope	1.066	1.033 to 1.101	<0.001	1.058	1.002 to 1.117	0.040
Peak VO_2_ at AT, mL/kg/min	0.819	0.687 to 0.978	0.027	0.831	0.551 to 1.256	0.380
O_2_ pulse, mL/kg/beat	0.839	0.706 to 0.996	0.045	0.956	0.775 to 1.181	0.678
Circulatory power, mmHg·mL/kg/min	0.999	0.998 to 1.000	0.002	0.999	0.997 to 1.000	0.131
Ventilatory power, mmHg	0.383	0.213 to 0.688	0.001	0.559	0.234 to 1.337	0.191
COP	1.106	1.045 to 1.170	<0.001	1.082	0.985 to 1.190	0.101
PetCO_2_ at rest, mmHg	0.850	0.771 to 0.937	0.001	0.866	0.749 to 1.000	0.050
PetCO_2_ at AT, mmHg	0.848	0.753 to 0.954	0.006	0.936	0.787 to 1.114	0.458

AT: Anaerobic threshold; BMI: Body mass index; COP: Cardiorespiratory optimal point; eGFR: Estimated glomerular filtration rate; LVEF: Left ventricular ejection fraction; PetCO_2_: Partial pressure of end-tidal carbon dioxide; pVO_2_: Peak oxygen consumption; VE/VCO_2_ slope: Minute ventilation-carbon dioxide production relationship.

**Table 4 biomedicines-11-02208-t004:** Area under the curve analysis for the composite endpoint in a 36-month follow-up.

	Group A (*n* = 283)			Group B (*n* = 204)			
CPET Parameters	AUC	95% CI	*p*-Value	AUC	95% CI	*p*-Value	*p*-Value * (Interaction)
pVO_2_, mL/kg/min	0.775	0.703–0.846	<0.001	0.732	0.608–0.856	0.003	0.571
Predicted pVO_2_ (%)	0.787	0.718–0.856	<0.001	0.725	0.594–0.855	0.004	0.420
VE/VCO_2_ slope	0.802	0.730–0.874	<0.001	0.752	0.601–0.903	0.003	0.569
pVO_2_, mL/kg/min at AT	0.569	0.253–0.885	0.711	0.682	0.475–0.888	0.070	0.719
O_2_ pulse, mL/kg/beat	0.629	0.540–0.719	0.003	0.670	0.517–0.823	0.029	0.478
Circulatory power, mmHg·mL/kg/min	0.779	0.713–0.845	<0.001	0.717	0.597–0.836	<0.001	0.913
Ventilatory power, mmHg	0.781	0.711–0.851	<0.001	0.719	0.587–0.852	0.001	0.915
COP	0.692	0.441–0.93	0.535	0.704	0.556–0.852	0.008	0.873
PetCO_2_ at rest, mmHg	0.599	0.515–0.684	0.025	0.716	0.584–0.848	0.003	0.095
PetCO_2_ at AT, mmHg	0.739	0.666–0.812	<0.001	0.705	0.512–0.899	0.024	0.567

* DeLong et al. *p*-values for comparison between groups. AT: Anaerobic threshold; COP: Cardiorespiratory optimal point; CPET: Cardiopulmonary exercise test; PetCO_2_: Partial pressure of end-tidal carbon dioxide; pVO_2_: Peak oxygen consumption; VE/VCO_2_ slope: Minute ventilation-carbon dioxide production relationship.

**Table 5 biomedicines-11-02208-t005:** Specificity and specificity of the cut-off values for the composite endpoint in a 36-month follow-up.

		Group A (*n* = 283)			Group B (*n* = 204)			
CPET Thresholds	Specificity	Sensitivity	Youden Index	PPV	Specificity	Sensitivity	Youden Index	PPV
pVO_2_ ≤ 12 mL/kg/min *	97%	25%	0.22	59%	76%	46%	0.22	13%
pVO_2_ ≤ 10 mL/kg/min	98%	13%	0.11	67%	94%	36%	0.30	29%
VE/VCO_2_ slope > 35	83%	58%	0.41	46%	52%	67%	0.19	10%
VE/VCO_2_ slope > 40	96%	35%	0.31	68%	81%	62%	0.43	20%
Percent of predicted pVO_2_ ≤ 50%	87%	51%	0.38	48%	60%	67%	0.27	12%
Percent of predicted pVO_2_ ≤ 40%	96%	24%	0.20	59%	88%	40%	0.28	20%

* pVO_2_ ≤ 12 mL/kg/min (or ≤14 mL/kg/min if intolerant of a β-blocker). pVO_2_: Peak oxygen consumption; VE/VCO_2_ slope: Minute ventilation-carbon dioxide production relationship. PPV: Positive predictive value.

## Data Availability

The data presented in this study are available upon request from the corresponding author. The data are not publicly available due to patient consent regarding the availability of individual patient data, applicable only to the local investigation team.

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
