# Peer review of "Cardiopulmonary Exercise Testing in the Age of New Heart Failure Therapies: Still a Powerful Tool?"

_biomedicines, 2023, doi:10.3390/biomedicines11082208_

Round 1

Reviewer 1 Report

attached you will find my comments.

only minor problems

Reviewer 2 Report

In the present study the Authors evaluated 487 HFrEF patients submitted to cardiopulmonary exercise testing (CPET) between 2009 and 2014 and between 2015 and 2018 confirming the prognostic value of CPET and specifically of peak oxygen consumption (pVO2) and VE/VCO2 slope but with different thresholds than the ISHLT recommended ones.

The main message of this paper is very important and very interesting.

However, I have major and minor concerns about this paper:

-        Patients in group B showed a lower pVO2, a higher VE/VCO2 and have more atrial fibrillation, suggesting a more severe disease; nonetheless, results showed a 7.4% vs 19.4% of Group A of the primary endpoint, namely an unbelievable demonstration of the power of the “new HF therapeutic approach”; please emphasize these unbelievable results at the beginning of the Discussion section.  

-        Several investigators stressed the desirability of adjusting the work rate increment according to the patient’s cardiorespiratory status showing that tests in which the incremental part of the protocol is completed in 6 and 12 minutes give the highest peak VO2 in normal subjects (Buchfuhrer et al. J Appl Physiol 1983; 55: 1558-1564). Wasserman et al. attempt to select a work rate increment that will result in termination of the incremental part of the exercise test in 8 to 10 minutes (Wasserman K et al. Principles of exercise test and interpretation; Lippincot Williams and Willis). Agostoni et al. (Eur J Heart Fail 2005; 7:498-504) described the same methodological concept to be true for heart failure patients too, confirming that the personalization of the protocol is mandatory to achieve results suitable for research trials, in particular at peak exercise. In the present paper, the authors used the same modified 96 Bruce protocol for all the patients with total durations out of the recommended ones (see above); moreover the duration of the tests are different between the two groups. Please, try to justify this methodological problem and/or cite it in the Limitation of the study session.

-        79% of the patients are males; please mention it in the Limitation of the study session

-        Several HF prognostic scores are validated; please mention them in the Discussion Section

-        To improve the easy reading of the Discussion Section, please build a table reporting different validated cutoff values for each variable

-        Paolillo et al. in 2019 published a paper on “HF prognosis over time: how the prognostic role of oxygen consumption and ventilatory efficiency during exercise has changed in the last 20 years” (European Journal of Heart Failure, 21 (2), pp. 208-217. DOI: 10.1002/ejhf.1364. Please discuss and cite it.

-        Cattadori et al. In 2013 published a paper on “Severe heart failure prognosis evaluation for transplant selection in the era of beta-blockers: Role of peak oxygen consumption” (International Journal of Cardiology, 168 (5), pp. 5078-5081. DOI: 10.1016/j.ijcard.2013.07.192. Please discuss and cite it.

Minor editing of English language required

Reviewer 3 Report

This study provides important and insightful information on the management of severe heart failure between the traditional drug era and the new Guideline-directed medical therapy (GDMT) era. In addition, this study has an appropriate methodology and understandable results. The CPET cut-off values used to determine the severity of heart failure have also been appropriately compared with historical data, providing information that could serve as a guiding policy for treatment guidelines in the new era. Study Limitation is also appropriately written, and although it is a single-center, retrospective study, it is thought to provide new insights in risk stratification for severe heart failure.

There are no comments. 

Round 2

Reviewer 1 Report

The authors have answered all questions which have been raised by the reviewer and thoroughly revised their manuscript. The reviewer does not have further comments.

excellent quality